# Formation of nitrogen-containing gas phase products from the heterogeneous (photo)reaction of NO$_2$ with gallic acid

Pan Li[1,2,3,4], Rachel Gemayel[5], Xue Li[6,7,8], Jiangping Liu[9], Mingjin Tang[1,2,3,4], Xinming Wang[1,2,3,4], Yan Yang[10,11,12 ✉], Hind A. Al-Abadleh [13 ✉] & Sasho Gligorovski [1,2,3 ✉]

Heterogeneous reaction of gas phase NO$_2$ with atmospheric humic-like substances (HULIS) is potentially an important source of volatile organic compounds (VOCs) including nitrogen (N)-containing compounds, a class of brown carbon of emerging importance. However, the role of ubiquitous water-soluble aerosol components in this multiphase chemistry, namely nitrate and iron ions, remains largely unexplored. Here, we used secondary electrospray ionization ultrahigh-resolution mass spectrometry for real-time measurements of VOCs formed during the heterogeneous reaction of gas phase NO$_2$ with a solution containing gallic acid (GA) as a proxy of HULIS at pH 5 relevant for moderately acidic aerosol particles. Results showed that the number of detected N-containing organic compounds largely increased from 4 during the NO$_2$ reaction with GA in the absence of nitrate and iron ions to 55 in the presence of nitrate and iron ions. The N-containing compounds have reduced nitrogen functional groups, namely amines, imines and imides. These results suggest that the number of N-containing compounds is significantly higher in deliquescent aerosol particles due to the influence of relatively higher ionic strength from nitrate ions and complexation/redox reactivity of iron cations compared to that in the dilute aqueous phase representative of cloud, fog, and rain water.

[1] State Key Laboratory of Organic Geochemistry and Guangdong Provincial Key Laboratory of Environmental Protection and Resources Utilization, Guangzhou Institute of Geochemistry, Chinese Academy of Sciences, Guangzhou 510640, China. [2] Guangdong-Hong Kong-Macao Joint Laboratory for Environmental Pollution and Control, Guangzhou Institute of Geochemistry, Chinese Academy of Science, Guangzhou 510640, China. [3] Chinese Academy of Science, Center for Excellence in Deep Earth Science, Guangzhou 510640, China. [4] University of Chinese Academy of Sciences, Beijing, China. [5] Institut National de l'Environnement industriel et des RISques (INERIS), Parc technologique Alata BP2, 60550 Verneuil en Halatte, France. [6] Institute of Mass Spectrometry and Atmospheric Environment, Jinan University, Guangzhou 510632, China. [7] Guangdong Provincial Engineering Research Center for On-line Source Apportionment System of Air Pollution, Guangzhou 510632, China. [8] Guangdong-Hongkong-Macau Joint Laboratory of Collaborative Innovation for Environmental Quality, Guangzhou 510632, China. [9] Faculty of Environmental Science and Engineering, Kunming University of Science and Technology, Kunming 650500, China. [10] School of Environmental Science and Engineering, Guangdong University of Technology, Guangzhou 510006 Guangdong, China. [11] Jieyang Branch of Chemistry and Chemical Engineering Guangdong Laboratory (Rongjiang Laboratory), Jieyang 515200, China. [12] Synergy Innovation Institute of GDUT, Shantou 515041 Guangdong, China. [13] Department of Chemistry and Biochemistry, Wilfrid Laurier University, Waterloo, ON N2L 3C5, Canada. ✉email: yangyan1209@gdut.edu.cn; halabadleh@wlu.ca; gligorovski@gig.ac.cn

Atmospheric humic-like substances (HULIS) originate from biomass burning and secondary chemistry[1,2] and comprise 48–57% of water-soluble organic compounds in aerosol particles[3] with concentrations varying from 0.08 μg m$^{-3}$ in a marine environment to 28 μg m$^{-3}$ in a rural area affected by forest fires[4,5]. HULIS is an important class of brown carbon (BrC), defined as sunlight absorbing organics capable of influencing the radiative forcing on regional and global scales[6]. Primary sources of BrC are mainly biomass burning[7]. Secondary sources of BrC from multiphase chemistry in the atmosphere[2] are an active area of ongoing research. For example, the oxidation of volatile organic compounds (VOCs) by atmospheric oxidants, especially in the presence of NO$_2$, would lead to the formation of BrC[1]. Given the importance of secondary organic aerosols (SOA) in affecting air quality, aerosol-cloud, and aerosol-radiation interactions in atmosphere[8], studies to date showed that the chemical diversity and physical properties of atmospheric aerosol particles influence the pathways that lead to BrC in SOA[9].

The heterogeneous reaction of HULIS with hydroxyl and nitrate radicals can promote the formation of SOA yielding the nitrated compounds[10]. Nitro-aromatic compounds are typically detected in aerosol particles[11]. Oxidation of HULIS by NO$_3$ leads to the formation of nitro-containing products, including nitrophenol, nitrated diphenyl, and nitro-products from non-aromatic/ring-opening[12]. Enhanced photosensitized uptake of NO$_2$ on wet humic acid thin films was studied and found to form HONO[13,14]. A recent study by Gen et al.[15] also reported the photogeneration of nitrite/HONO from the photolysis of iron–organic complexes in the presence of nitrate. However, other gas and condensed phase BrC products from the heterogeneous reaction of HULIS with NO$_2$ have yet to be identified, particularly under experimental conditions that simulates the complex physicochemical properties of atmospheric aerosol particles of liquid water content, pH, ionic strength, and presence of reactive transition metals such as iron, Fe.

Mineral dust is a dominant source of Fe in atmospheric aerosols in addition to combustion sources[16–19]. During biomass burning events, analysis of the particles showed that metals such as Fe coexist with organic compounds[20]. Also, processing of mineral dust particles takes place in the atmosphere through reactions with acidic gases, organics, in the dark or under irradiation, changing the hygroscopicity of these particles and increasing the amount of dissolved Fe. The concentrations of Fe in raindrops, fog, and cloud waters range from 0.1 to 1138 × 10$^{-6}$ mol L$^{-1}$ [21] depending on the location with typical concentration in cloud droplets of about 10$^{-6}$ mol L$^{-1}$ [21]. These processed Fe-containing particles can act as a sink for semivolatile organics from biomass burning and undergo mixing with biomass burning organic aerosol. The role of Fe in the aqueous phase formation of water-soluble and insoluble secondary light-absorbing organic products was reported in a number of lab studies using standard organic compounds[22–24] or acidic extracts of biomass burning organic aerosol[24]. These studies highlighted the effect of Fe-catalyzed reactions on the morphology, hygroscopic, and ice nucleation properties of insoluble products and standard dust samples.

The objective of this investigation is to identify gas phase reaction products from the heterogenous reaction of gas phase NO$_2$ with GA solutions containing nitrate and Fe ions, in the dark and under irradiation (300 nm < λ < 700 nm). We chose GA as a polyphenolic model of HULIS because the chemical structure of GA (Fig. 1, p$K_{a1}$ = 4.16, p$K_{a2}$ = 8.55, p$K_{a3}$ = 11.4, p$K_{a4}$ = 12.8)[25] has the functional groups found in real HULIS[26] and is more representative than the other aromatic proxy compounds such as catechol, syringol, guaiacol, gentisic acid, and resorcinol shown in Fig. 1[27–30]. Nitrate (NO$_3^-$) is the most dominant water-soluble ion in the pH from 3 to 6[31], and Fe(III) is the most ubiquitous metal in the cloud, rain, fog with the pH ranging from 2 to 7 under oxic conditions[21]. Hence, aqueous solutions consisting of GA, GA/NO$_3^-$, GA/Fe(III), and GA/NO$_3^-$/Fe(III) were prepared as described in the methods section.

Real-time measurements of the gas phase products were performed using secondary electrospray ionization ultrahigh-resolution mass spectrometry (SESI-UHR-MS)[32–34] under positive and negative ionization modes. The technique we used is state of the art mass spectrometry that provides qualitative information on the products (masses and hence structural information) and quantitative information from the relative intensities. The results indicate that the presence of NO$_3^-$ and Fe(III) leads to an enhanced number of product compounds formed during the NO$_2$ reactions with GA, especially under dark conditions. An important fraction of N-containing organic compounds were detected in the presence of NO$_3^-$ and Fe(III). Intriguingly, the increase of the amount of NO$_3^-$ in the presence of Fe(III) led to reduced number of N-containing product compounds. Majority of the produced N-containing compounds have reduced N-functional groups such as the polyamine compounds (putrescine), and aromatic amines containing two phenyl substituents, such as 4-dimethylaminobenzaldehyde, which could be toxic compounds.

## Results and discussion

**Characterization of the gas-phase product compounds.** The formation of gas phase products was directly monitored using SESI-UHR-MS at a time resolution of 1 s. The data of gas products are shown in the Supplementary Data[35]. Figure 2A shows a hierarchical cluster analysis of 765 ions with intensities above 1 × 10$^5$ au captured in real-time measurements upon heterogeneous reaction of gaseous NO$_2$ with aqueous GA/NO$_3^-$/Fe(III) in the dark. After removing the isotope peaks and conducting a statistical significance analysis test (two-tailed test, p < 0.0001), 53 ions shown in the group marked with a red rectangle in Fig. 2A were distinguished from all the monitored gas-phase compounds. The ions in the red rectangle exhibited an increasing intensity trend as the solution containing GA/NO$_3^-$/Fe(III) was exposed to gas phase NO$_2$ in dark, which suggests that the detected ions are compounds formed during the NO$_2$ reaction.

Figure 2B shows the results of the two-tailed test of m/z at 91.9981, 103.0387, 117.0544, and 131.0702. These results highlight statistically significant difference between the intensities at the zero-air stage and during the NO$_2$ flow into the GA/NO$_3^-$/Fe(III) solution indicating that these ions are product compounds. Figure 2C shows the average mass spectra of the detected gas phase compounds, which include the relative number abundance. The intensities of the ions in the measured samples were normalized to the most abundant peak C$_7$H$_3$O$_5$N (m/z 182.0090) ion as a reference compound (100%). The identified formulas are classified into three major compound categories as CHO$^+$, CHON$^+$, CHN$^+$ according to the formula identification by means of Xcalibur software. Out of the 53 ions detected, the most abundant are CHON$^+$ (49.1%) followed by CHO$^+$ (36.8%), CHN$^+$ (7%), and 7% are unassigned compounds. Figure 2D shows temporal resolution of m/z at 91.9981, 103.0387, 117.0544, and 131.0702 formed upon reaction of NO$_2$ with GA in the presence of NO$_3^-$ and Fe(III).

The influence of NO$_3^-$ and Fe(III) on the formed gas phase product compounds during the heterogeneous reaction of NO$_2$ with GA shown in Fig. 2 can be better explained by the Van Krevelen (VK) plots depicted in Fig. 3. Namely, the number of oxidized product compounds largely increased upon the addition of [NO$_3^-$] = 0.05 mol L$^{-1}$ and [Fe(III)] = 1 × 10$^{-6}$ mol L$^{-1}$

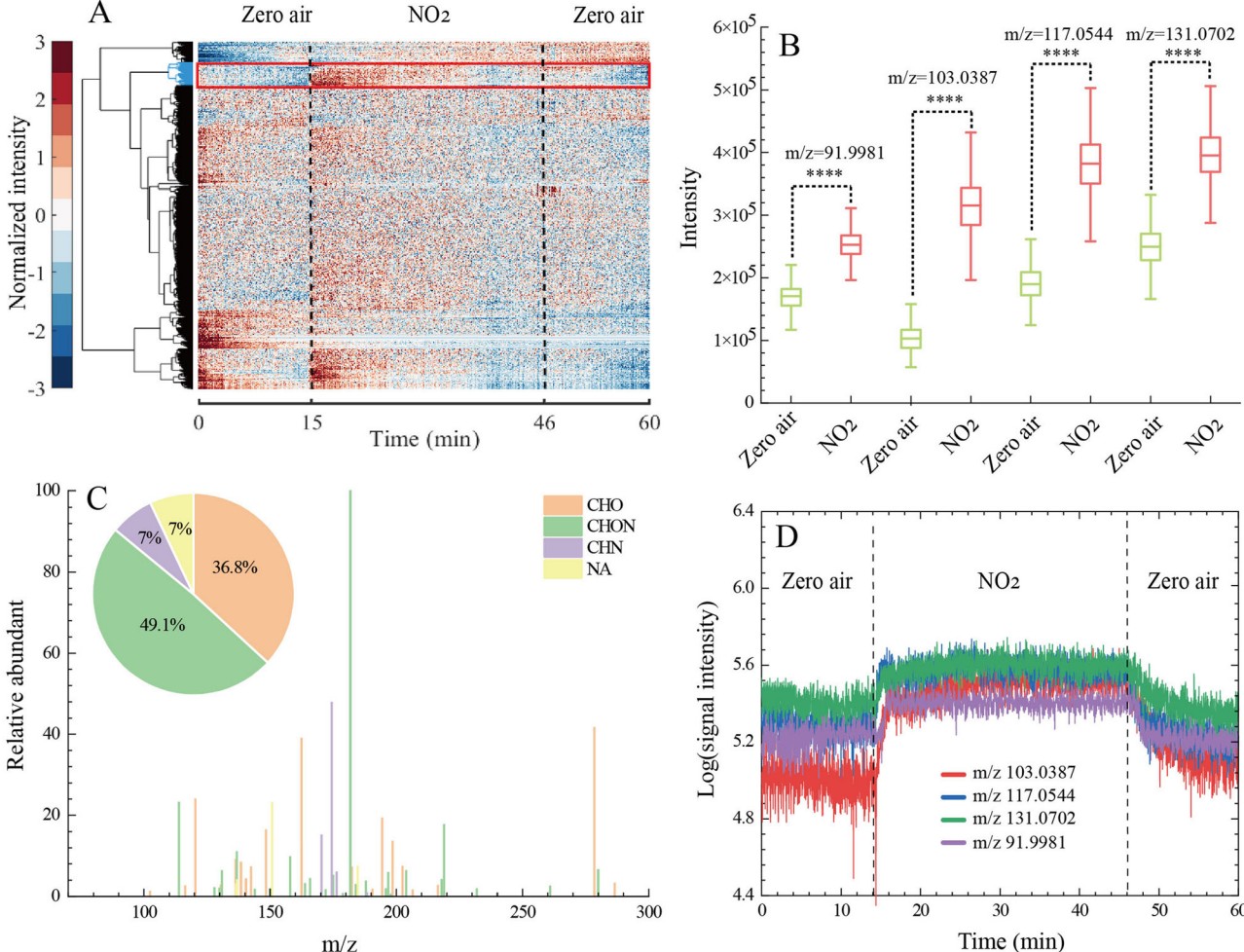

**Fig. 1 Chemical structure of the compounds.** Selected standard chemical compounds used in lab studies as models for HULIS.

**Fig. 2 Gas phase product analysis from the heterogeneous reaction of gas phase NO$_2$ with solutions of GA. A** Hierarchical cluster analysis of 765 ions detected in ESI+ mode upon heterogeneous reaction of NO$_2$ with GA/Fe(III)/NO$_3^-$ ([GA] = 1×10$^{-6}$ mol L$^{-1}$, [NO$_3^-$] = 0.05 mol L$^{-1}$, [Fe(III)] = 1×10$^{-6}$ mol L$^{-1}$) in the dark. The normalized signal intensity in the hierarchical cluster is presented by a color-coded scale; i.e., the signal intensity increases from dark blue (normalized value of −3) to wine red (normalized value of 3). **B** Paired sample $t$ test showing a significant difference (****$p < 0.0001$) in the concentration of each ion at the NO$_2$ stage and zero-air stage in the dark. **C** A mass spectrum of the observed ions in ESI+ mode. Different formula groups are color-coded. The pie chart shows the proportion of CHO$^+$, CHON$^+$, CHN$^+$, and unassigned peaks among all observed product compounds in the dark. **D** Real-time formation profiles of four product compounds that have been detected during the heterogeneous reaction of NO$_2$ with GA/NO$_3^-$/Fe(III) in the dark. All of the data displayed here correspond to the products detected in ESI+ mode.

especially during the reaction of NO$_2$ with GA in dark (Fig. 3C, D).

Intriguingly, data in Supplementary Fig. S1 show that the increase of the amount of nitrate ions from 0.05 to 0.5 mol L$^{-1}$ led to reduced number of the formed gas phase oxidized compounds during the reaction of NO$_2$ with GA/Fe(III) in dark, while there is no significant difference under light irradiation.

Figure 3C shows that a majority of the gas phase product compounds are located in the region with 1 < H/C < 2 and O/C < 1, which is consistent with compounds identified in ambient aerosols (Supplementary Table S1)[33,36]. Figure 2C also shows that CHO compounds (44.7%) depicted in the range between H/C ≥ 1.5 and O/C ≤ 0.75 are most likely aliphatic compounds (alkanes, alkenes, and cyclic compounds)[36–38]. The compounds with Double Bond Equivalency (*DBE*) values ranging from 6 to 11 located in the domain with H/C ratio ≤1 and O/C ratio ≤0.5 in Fig. 2C indicate that they are polyaromatic compounds and unsaturated aromatic compounds, respectively. The compounds shown in Fig. 3C are classified in 6 CHO series compounds (Supplementary Fig. S2), exhibiting the same homologous CH$_2$

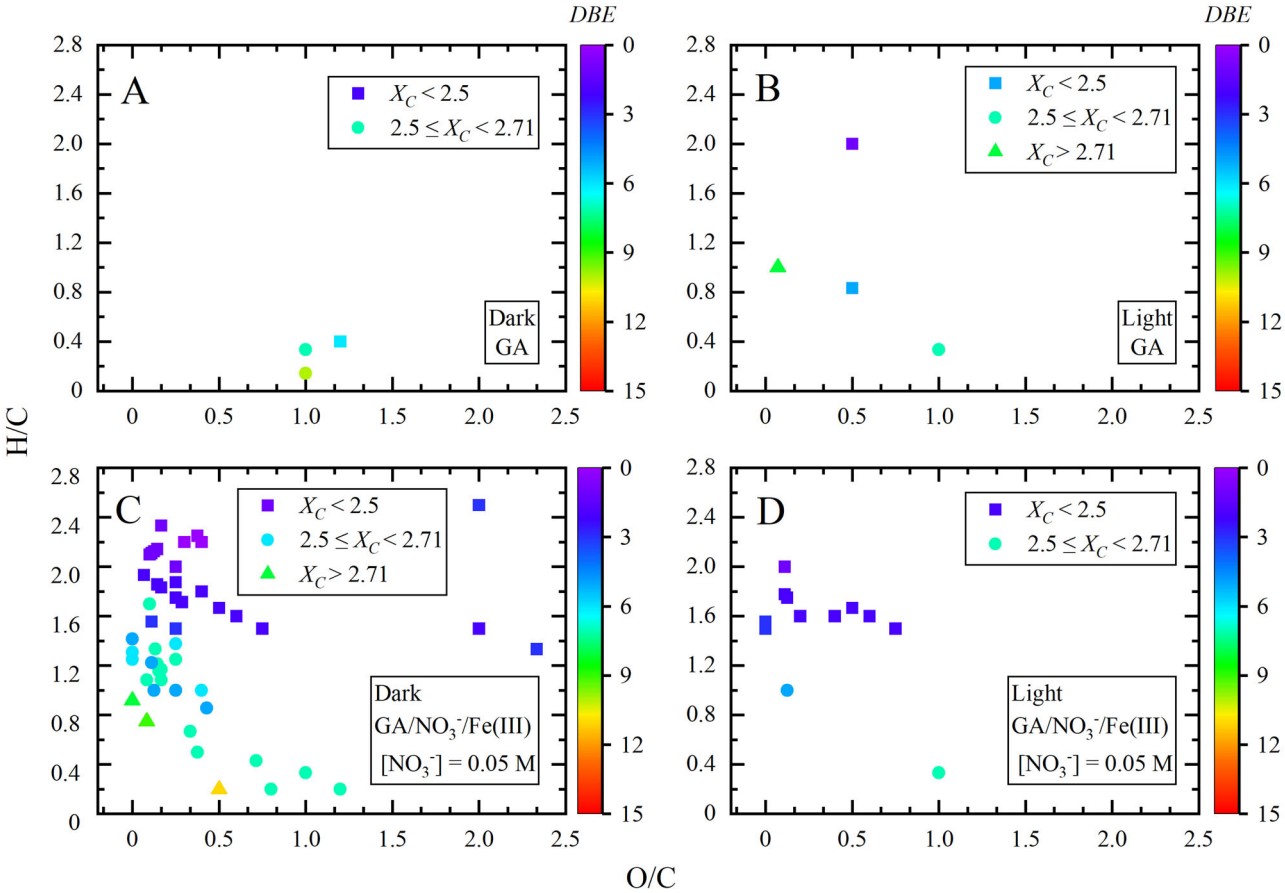

**Fig. 3 Van Krevelen plot showing the correlation between H/C ratio and O/C ratio obtained for the organic compounds detected in heterogeneous reaction of $NO_2$ with GA, and GA/$NO_3^-$/Fe(III) ([$NO_3^-$] = 0.05 mol L$^{-1}$). A** GA in dark, **B** GA under light irradiation, **C** GA/$NO_3^-$/Fe(III) ([$NO_3^-$] = 0.05 mol L$^{-1}$) in dark, and **D** GA/$NO_3^-$/Fe(III) ([$NO_3^-$] = 0.05 mol L$^{-1}$) under light irradiation. *DBE* refers to Double Bond Equivalency.

series. For example, Supplementary Fig. S2 shows the series of $C_8H_{18}O_3(CH)_2$, $C_5H_8O_3(CH)_2$, $C_7H_{12}O_2(CH)_2$, $C_{10}H_{18}O_4(CH)_2$, with $DBE < 2$ and $X_C < 2.5$, which could be the aliphatic compounds[36]. The series of $C_{10}H_{10}O_4(CH)_2$ and $C_8H_4O_3(CH)_2$, exhibiting the range in *DBE* values from 6 to 7 and the $2.5 < X_C < 2.7$, are monoaromatic compounds (Supplementary Fig. S2)[36].

In addition to the formed CHO compounds, a large fraction of N-containing compounds (55.3%) were formed during the heterogeneous reaction of $NO_2$ with GA in the presence of $NO_3^-$ ([$NO_3^-$] = 0.05 mol L$^{-1}$) and Fe(III) in the dark (Supplementary Table S1). Supplementary Tables S2–S12 show the products compounds detected under different conditions applied in this study. Figure 4 shows a VK plot obtained for the N-containing compounds in the form of H/C versus N/C, illustrating different ions with the same homologous $CH_2$ series. The upper left corner of Fig. 4 in the range between H/C > 2 and N/C < 0.4 shows series of $C_7H_{15}ON(CH)_2$ compounds with one N atom and $DBE = 1$, which are associated with aliphatic amines[39]. The series of $C_{12}H_{14}O_2N_2(CH)_2$, $C_{12}H_{15}N(CH)_2$, and $C_{12}H_{13}O_2N(CH)_2$ have *DBE* values ranging from 6 to 11, implying the presence of N-heterocyclic compounds with condensed aromatic rings (Fig. 4)[40]. The VK plot in Fig. 4 for H/C versus N/C shows that there are four CHN compounds with the *DBE* values ranging from 5 to 8. Three of the CHN compounds, with *DBE* values from 5 to 6 and $X_C > 2.6$, are monoaromatic compounds ($2.5 \leq X_C < 2.7$) and one is polyaromatic compound ($X_C > 2.7$) with $DBE = 8$ (Fig. 4 and Supplementary Table S1). The CHN

compounds with six-membered rings are N-cyclohexylaniline ($C_{12}H_{17}N$), n-phenylcyclohexaneimine ($C_{12}H_{15}N$), and diphenylamine ($C_{12}H_{11}N$).

Moreover, gas phase compounds detected in the ESI+ ion mode are divided into different subgroups according to the O/N ratios. Figure 5 shows the contribution of each subgroup to the total

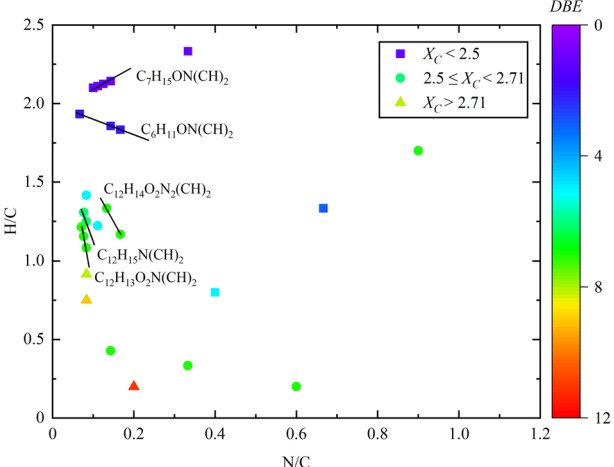

**Fig. 4 Van Krevelen plot showing the correlation between H/C ratio and N/C ratio.** The N-containing compounds detected in heterogeneous reaction of $NO_2$ with GA/Fe(III)/$NO_3^-$ ([$NO_3^-$] = 0.05 mol L$^{-1}$) in the dark.

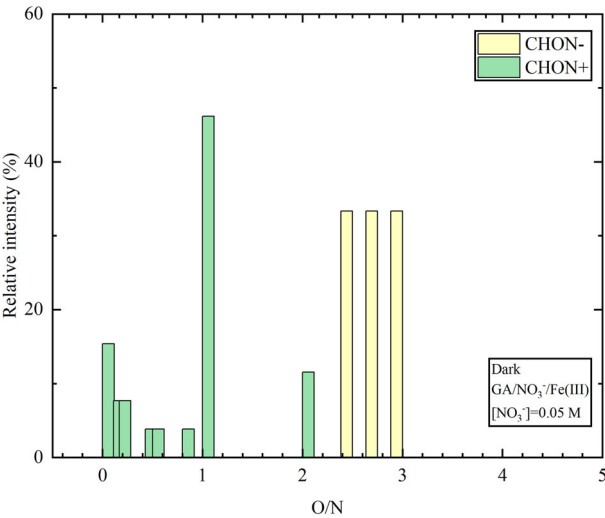

**Fig. 5 N-containing organic compounds divided into subgroups according to the O/N ratio in their chemical composition.** The y-axis represents the contribution of each subgroup to the total intensity of CHON compounds in ESI+ and ESI− ion modes detected during the heterogeneous reaction of $NO_2$ with GA in dark, in the presence of $NO_3^-$ ([$NO_3^-$] = 0.05 mol L$^{-1}$) and Fe(III).

intensity of CHON compounds in ESI+ ion mode. About 96% of the detected N-containing compounds exhibit O/N ratio ≤2. This result is reasonable considering that these compounds have reduced N-containing functional groups (e.g., amines, imines, and imides), which are easily protonated in ESI+ ion mode[41].

Intriguingly, a majority (48%) of CHON compounds have reduced N-containing functional groups with O/N ratio of 1, such as $C_6H_{11}ON$, and heptanamide ($C_7H_{15}ON$), which are consistent with compounds identified in ambient aerosols[42]. CHON compounds with O/N ratio of 0.5 indicate the presence of a polyamine (1,4-diaminobutane) ($C_6H_{14}ON_2$), which was also detected in atmospheric aerosols[43]. The gas phase compounds with C=N bond such as 1,4-diaminobutane (putrescine) and aromatic amines with two phenyl substituents, (4-dimethylaminobenzaldehyde), could be toxic compounds[44]. Another interesting subgroup is the one with O/N ratio of 0, which accounts for 16% of the total number of the detected N-containing organic compounds[41]. The compounds in this subgroup (O/N = 0) include $C_{12}H_{17}N$, $C_{13}H_{17}N$, and $C_{12}H_{11}N$, which have been also detected in ambient aerosols[36].

It has to be noted that compounds with O/N ≤ 3 refer to reduced N-containing functional groups with C=N and C−N groups in their structure. These bonds were identified as the two most important chemical bonds of BrC in aerosol samples collected in Shanghai[45]. Examples of these compounds of atmospheric relevance include $C_{22}H_{43}NO$, $C_{14}H_{12}N_4O_4$, $C_{16}H_{15}N_3O_5$, $C_{24}H_{41}NO$, $C_{27}H_{41}NO$, and $C_{35}H_{29}N_5O_2$[45]. Supplementary Figs. S3, S4, and S5 show that a small fraction of the N-containing product compounds contains oxidized nitrogen ($CH_3O_4N$, and $C_{10}H_4O_8N_2$) as detected in ESI− mode. These compounds with $X_C > 2.5$ exhibit O/N = 4 indicating they are aromatic compounds which contain nitrooxy (−$ONO_2$) group; hence, they could be light-absorbing compounds associated with BrC in the atmosphere[36,46]. It has to be noted, however, that the same chemical formula does not necessarily imply that it is the same molecular structure as number of structural isomers are plausible for each formula[47].

**Role of the nitrate ion**. The $NO_3^-$ ion was added to the GA solutions prepared here to adjust the ionic strength at two concentrations, 0.05 and 0.5 mol L$^{-1}$, which are 500–5000× higher than GA concentration. The ionic strength can affect the kinetics and product distribution within the aerosol deliquescent particles[48,49]. It has been shown that the addition of $NO_3^-$ substantially favors the formation of polyaromatic compounds and substances with nitrooxy or oxygenated nitrooxy group of organonitrates through heterogeneous ozone processing of methoxyphenols[46]. Loisel et al.[50] have shown that the photolytic degradation of vanillin in the presence of $NO_3^-$ in the aerosol liquid water leads to the formation of condensed aromatic N-containing compounds compared to the corresponding degradation in the dilute aqueous phase of cloud droplets, which resulted in the formation of aliphatic-like CHO compounds. Degradation rates of syringaldehyde and acetosyringone in aqueous solutions containing $NO_3^-$ were 1.5 and 3.5× faster than corresponding degradation rates in sulfate solutions[51]. It seemed that $NO_3^-$ promoted the degradation of syringaldehyde and acetosyringone because nitrate ions are a photolytic source of OH radicals, while sulfate is not, highlighting the importance of $NO_3^-$ in the formation of SOA by facilitating the photolytic degradation of organic compounds[51].

In the dark, and under the experimental conditions described here, spontaneous redox reactions can take place driven by two-electron transfer between $NO_3^-$ and GA. This is because the electrochemical reduction of $NO_3^-$ to $HNO_2$, $N_2O_4$, and $NO_2^-$ under acidic to neutral conditions requires two electrons[52] according to the following Equations:

$$NO_3^- + 3H^+ + 2e \rightleftarrows HNO_2 + H_2O, \; E_{red}^\circ = 0.934 \, V \quad (1)$$

$$2NO_3^- + 4H^+ + 2e \rightleftarrows N_2O_4 + 2H_2O, \; E_{red}^\circ = 0.803 \, V \quad (2)$$

$$NO_3^- + H_2O + 2e \rightleftarrows NO_2^- + 2OH^-, \; E_{red}^\circ = 0.01 \, V \quad (3)$$

The electrochemical oxidation of GA was studied as a function of pH and found to occur via an irreversible oxidation of GA to the semiquinone radical cation (GA$^{\bullet+}$) by an irreversible one-electron transfer process[53]. Then, the formed radical cation loses a proton to form the semiquinone radical (GA$^\bullet$) followed by a second irreversible one-electron transfer to the quinone cation (GA$^+$). Finally, deprotonation of GA$^+$ to give the quinone (GAO) completes the overall two-electron process[53]. At pH 5, the oxidation potentials of the first and second electron transfer steps were calculated to be 0.445 and 0.741 V[53]. Both of these oxidation potentials are lower than the reduction potentials of $NO_3^-$ in Eqs. ((1)–(3)), which results in a net positive electrochemical cell potential, hence thermodynamically favorable redox paired reactions. In addition, data in Figs. 1C and S1 show that increasing nitrate concentrations under dark conditions significantly alter the number of gaseous products. Since GA is the limiting reagent in these redox paired reactions, it is very likely that the increase in ionic strength from increasing the nitrate concentration under dark conditions dramatically reduces the kinetics of subsequent oxidation pathways of GA following its initial oxidation to quinones. As a results, a limited number of products with $X_C < 2.5$ are observed at [$NO_3^-$] = 0.5 M compared to the larger number of products with $2.5 \leq X_C < 2.71$ and $X_C > 2.71$ at [$NO_3^-$] = 0.05 M.

Under light irradiation, the absorption spectrum of $NO_3^-$ is dominant by a weak n → π* band at 302 nm ($\varepsilon$ = 7.2 M$^{-1}$ cm$^{-1}$) implying that nitrate ions can be photolyzed upon light irradiation of aqueous solutions containing nitrate ions producing oxidants such as hydroxyl radical (OH) and $NO_2$, and nitrite ions

($NO_2^-$) as shown below[54–58]:

$$NO_{3(aq)}^- \xrightarrow{hv} NO_{2(aq)}^- + OH \qquad (4)$$

$$NO_3^- + hv + H^+ \rightarrow NO_2 + OH \qquad (5)$$

Considering that $pK_{a1}$ of GA is 4.16[25], a fraction of the compound is dissociated, at pH 5, implying fast photodegradation[59]. As a phenolic acid, GA is a known powerful natural antioxidant[60]. Its relatively low oxidation potential under acidic to neutral pH with three hydroxyl substituents on the benzene ring explain its strong free radical scavenging activity[61]. It has been shown that the reaction of GA with OH proceeds very fast with the rate constant $(1.1 \pm 0.1) \times 10^{10}\,M^{-1}\,s^{-1}$ at pH 6.8[59,62,63], which is close to the diffusion limit[64]. The reaction of OH with GA at pH 6.8 leads to the formation of an adduct (GA-OH) which further reacts with GA yielding phenoxyl type radicals[63].

It has been shown that the reaction of $NO_2$ with HULIS can possibly lead to the formation of nitrous acid (HONO)[13]. For example, the heterogeneous reaction of $NO_2$ with dry humic acid (prepared in aqueous solution, $1\,mg\,mL^{-1}$, pH = 4.4) under light irradiation produced high amounts of gaseous HONO[13]. However, it has been shown that HONO cannot be detected in the gas phase during the heterogeneous reaction of $NO_2$ with resorcinol used as a proxy of HULIS, at pH 6 or above, because all nitrite ions ($NO_2^-$) remain in the liquid film, which is reasonable considering that the $pK_a$ for HONO is 3.5[30]. In our studies, the initial pH value of 5 increased in the course of the reaction to 11 (Supplementary Table S13), indicating that it is unlikely that HONO could be formed during the reaction of $NO_2$ with GA. Nevertheless, we have observed a small amount of HONO in the gas phase ranging between 0.3 and 0.5 ppb during the reaction of $NO_2$ with GA in presence of $NO_3^-$ ([$NO_3^-$] = 0.05 mol $L^{-1}$) in dark and under irradiation, respectively (Supplementary Fig. S6).

**Role of Fe(III).** Considering that GA exists in excess of Fe(III) by a factor of 100, implies that Fe(III) will be complexed by three GA molecules in the dark, which will suppress the oxidation of GA in dark with and without nitrate ions because $GA_3Fe$ complexes are stable[65]. Under irradiation, and in the absence of nitrate, which means no OH radicals, the known reactions involve Fe(III) through the formation of a phenoxy radical intermediate, followed by reaction with OH + $NO_2$, are described by the following set of Eqs. ((6)–(10))[27,66]:

$$[Fe(III) - GA]^{2+} \xrightarrow{hv} Fe(II) + GA \qquad (6)$$

$$GA \cdot + NO_2 \rightarrow GA{-}NO_2 \qquad (7)$$

$$GA{-}NO_2 \cdot + NO_2 \rightarrow HONO + GA{-}NO_2 \qquad (8)$$

$$GA - NO_2 \cdot + NO_2 \rightarrow GA(NO_2)_2 \qquad (9)$$

$$2GANO_2 \cdot \rightarrow \text{recombination products} \qquad (10)$$

Light absorption by [Fe(III)-GA]$^{2+}$ leads to the formation of Fe(II) and phenoxy radicals (GA·) Eq. (6)[27,67], followed by the reaction of phenoxy radicals with $NO_2$ to yield the recombination products ((7) to (10))[66].

The influence of Fe(III) on the HONO formation upon reaction of $NO_2$ with GA in dark and under light irradiation is shown in Supplementary Figs. S7, S8. In the dark, about 0.3 ppb of HONO is formed, which increases up to 0.7 ppb under irradiation. The conversion of $NO_2$ to HONO could be facilitated by Fe(III)[15]. Also, the complexed GA-Fe(III) complexes would be capable of protonating nitrite *via* hydrogen abstraction by $NO_2$

due to the presence of hydroxyl groups on the benzene ring of GA and hydroxyl ligands on complexed Fe(III) to GA. Therefore, although the GA is responsible for reducing $NO_2$ to $NO_2^-$, the presence of Fe(III) can facilitate the protonation of $NO_2$ and the release of gaseous HONO even at pH 5 that is higher than the $pK_a$ of HONO ($pK_a = 2.8$)[68]. The amount of HONO formed from aqueous GA mixed with Fe(III) shown in Supplementary Fig. S7 is slightly higher than that mixed with $NO_3^-$ (Supplementary Fig. S6), indicating that indeed Fe(III) ions are capable of protonating nitrite.

Under irradiation, the HONO enhancement compared to dark observed in Supplementary Fig. S7 could be ascribed to photoexcitation of GA, followed by heterogeneous reduction of $NO_2$ to HONO[13,69–71]. Substrates like GA that act as photo-sensitizing or photoreducing agents demonstrate this effect to the greatest extent[69]. This reduction has been proven to occur on organic substrates including humic substances, soil[13], and urban grime[72,73], at wavelengths not only in the UV-A spectral region, but also in the visible region under atmospheric conditions[13]. It has been shown that $NO_2$ may be more efficiently hydrolyzed in aqueous solution when it is generated in situ during nitrate photolysis than during the heterogeneous reaction where mass transfer of gaseous $NO_2$ into bulk solution is too slow[71].

Benzoquinones are photoactive, which can generate semiquinone radicals when excited by light and in the presence of H-donors[74]. Therefore, it is reasonable to hypothesize that GA can reduce $NO_2$ by the quinones or semiquinone species generated by Fe(III) chemistry in the dark or during light irradiation (Supplementary Fig. S11), followed by the conversion of $NO_2$ to HONO in dark as well as during the irradiation.

The amounts of HONO are consistent with the mechanism, where the reduced species such as hydroquinone and semiquinones are generated when the quinones (forming from GA oxidation) are excited by light (Supplementary Fig. S12). Following their reaction with $NO_2$, the phenols are reoxidized to the quinone state[75], which can be cycled back to a phenol upon absorption of a photon or a chemical reductant. This mechanism accounts for difference in reactivity shown in Supplementary Fig. S11 and Supplementary Fig. S12. The efficiency of the reaction between $NO_2$ and GA in dark decreases over time as semiquinone and hydroquinone reactive sites are oxidized to quinones (see Supplementary Fig. S11)[68]. In contrast, the photochemical reactivity observed in Supplementary Fig. S12 is relatively constant over the course of the experiment because photochemistry provides a mechanism which allows to regenerate semiquinones and hydroquinone reactive sites[68].

The N-containing compounds formed upon heterogeneous reaction of $NO_2$ with GA in the presence of Fe(III) in dark can be tentatively explained by formation of $GA_3Fe$ complexes ($A_1$)[65], which reacts with $NO_2$ to generate Fe(II) and phenoxy radical ($A_2$). $A_2$ obtains an electron and an H atom to form phenoxyl radicals ($A_3$)[68], which can abstract H atom from hydroxy group of GA (Supplementary Fig. S13). Then $A_3$ undergoes nucleophilic substitution reaction in the presence of $NO_3^-$ to yield 3,4-dihydroxy-5-nitrobenzoic acid ($C_7H_5O_6N$) ($A_4$)[46,76,77]. $A_4$ can be reduced in the presence of Fe(III) to 5-amino-3,4-hydroxybenzoic acid ($C_7H_7O_4N$) ($A_5$)[78–80]. $A_5$ can abstract H atom from hydroxy group to form a radical, followed by the nucleophilic substitution reaction to generate 4-hydroxy-5-amino-3-nitrobenzoic acid ($C_7H_6O_5N_2$) ($A_7$)[46,76,77,81]. $A_7$ can be reduced in the presence of Fe(III) to 4-hydroxy-2,5-diaminobenzoic acid ($C_7H_8O_3N_2$) ($A_8$)[78–80]. The $A_8$ could abstract H atom from hydroxy group by $NO_2$ radical to yield 2,5-diamino-1,4-benzoquinone ($C_6H_6O_2N_2$) ($A_9$)[82]. Finally, the $A_9$ could be oxidized by OH radical to form ($C_4H_6O_3N_2$) ($A_{10}$)[83], which undergoes decarboxylation leading to putrescine ($C_4H_{12}N_2$) ($A_{11}$)[84] (Supplementary Fig. S13).

**Conclusions**. We showed that $NO_2$ heterogeneous (photo)reactions of GA used as a proxy of HULIS in the presence of $NO_3^-$ and Fe(III) ions can lead to the formation of a broad range of gas phase unsaturated and saturated organic compounds under atmospherically-relevant conditions. These results suggest that heterogeneous uptake of gaseous $NO_2$ to wet aerosols containing HULIS would produce relatively higher amounts of VOCs than in the cloud droplets, considering that the ionic strength in deliquescent aerosol particles is much higher compared to the one in dilute aqueous phase in rain, fog, and cloud waters[48,85]. The majority of the formed compounds contain N atom in their structure, which can have a detrimental impact on human health upon inhalation and the environment given their toxicity and unique chemical reactivity. For example, the polyamine compounds, such as 1,4-diaminobutane (putrescine), and aromatic amines containing phenyl substituents, such as 4-dimethylaminobenzaldehyde, are potentially toxic compounds[44,86]. The secondarily formed amines, can be further nitrosated in the atmosphere generating N-nitroso compounds such as nitrosamines known for their adverse health effects[87]. In addition, the N-containing VOCs formed through the $NO_2$ reactions with HULIS investigated here can have an additional contribution to the formation of light-absorbing SOA[88,89].

This study forms the basis for future experimental work designed to obtain additional quantitative information such as absolute amounts of selected products and kinetics of reaction steps in the mechanism under the experimental conditions reported here. While bubbling $NO_2$ gas into an aqueous phase solution followed by analyzing gas phase products is not identical to flowing $NO_2$ over an aerosol particle or a suspended microdroplet, these experimental setups share the same elementary steps in the reaction mechanism such as adsorption, surface diffusion, accommodation to the bulk, and bulk diffusion[90]. Hence, results from the experimental setup described here provide a reference point and a bench mark for future work that aims to examine the roles of surface to volume ratio, particle curvature, and aerosol liquid water acidity and ionic strength.

## Methods

**Chemicals**. Gallic acid (GA) (≥98.0%, CAS: 149-91-7, Coolaber), iron(III) chloride hexahydrate ($FeCl_3 \cdot 6H_2O$) (≥98.0%, CAS: 10025-77-1, Aladdin), nitrate ($NaNO_3$) (≥99.0%, CAS: 14797-55-8, Sigma Aldrich), sodium hydroxide (NaOH) (≥96.0%, CAS: 1310-73-2, Aladdin), hydrochloric acid (HCl) (ACS reagent grade, 99%, CAS: 7647-01-0, Guangzhou Chemical Reagent).

**Experimental conditions**. A double-well rectangular (5 cm × 5 cm × 2 cm) reactor made up of borosilicate glass was used to evaluate the formation of condensed phase products compounds and compounds released in the gas phase in dark and under light irradiation (Supplementary Fig. S9)[32]. A mixing ratio of 50 ppb obtained by diluting a pure $NO_2$ flow (4.1 mL min$^{-1}$) (0–10 mL min$^{-1}$ mass flow controller, Seven Star, China) from a certified $NO_2$ bottle gas of 10 ppm with air flow (800 mL min$^{-1}$) (0–1000 mL min$^{-1}$, Seven Star, China) was continuously passing through the reactor during the experiments. $NO_2$ concentrations were measured online at the exit of the flow tube using an $NO_2$ analyzer (Thermo Scientific, model 42i). The detection limit of the $NO_x$ analyzer was 300 ppt with a time resolution of 30 s. The reactor was filled with 10 mL of freshly prepared solution and exposed to $NO_2$ in dark and light irradiation by a xenon lamp (Xe, 500 W, 300 nm < λ < 700 nm). The temperature in the flow tube reactor was held constant at 293 ± 0.02 K by water circulating through its jacket, which was connected with a thermostated bath (Lauda RC, Germany). Aqueous solutions consisting

of GA ([GA] = $1 \times 10^{-4}$ mol L$^{-1}$), or of GA mixed with $FeCl_3 \cdot 6H_2O$ ([Fe(III) = $1 \times 10^{-6}$ mol L$^{-1}$]), or of GA mixed $NaNO_3$ ([$NO_3^-$] = 0.05, 0.5 mol L$^{-1}$) or of GA mixed with $FeCl_3 \cdot 6H_2O$ and $NaNO_3$ ([$NO_3^-$] = 0.05, 0.5 mol L$^{-1}$), were prepared with ultra-pure water (Sartorius 18 MΩ cm, H₂O-MM-UV-T, Germany). The pH values of these solutions were monitored by a pH meter (Mettler Toledo). To simulate the pH of clouds, fog and moderately acidic particles, pH was adjusted to 5 using sodium hydroxide (NaOH) ([NaOH] = 1 mol L$^{-1}$) and hydrochloric acid (HCl) ([HCl] = 0.1 mol L$^{-1}$) diluted from 37% v/v (ACS reagent grade).

**Sample analysis using SESI-UHR-MS**. The real-time measurements of gas phase compounds were performed by a SESI-UHR-MS (FIT, Spain; Q-Exactive, Thermo Scientific, USA) in both negative (ESI-) and positive (ESI+) ionization modes. In addition, blank measurements were performed before and after analysis. A SESI-UHR-MS and an $NO_2$ analyzer were coupled to the double-wall rectangular reactor to observe the formed gas-phase products and the $NO_2$, respectively, during the whole measurement period. The mass analyses ranged from 50 to 500 m/z, with a resolution above $1 \times 10^5$ arbitrary units (au) and a mass accuracy ≤2 ppm[32,34]. Regarding the experiments performed in dark and under light irradiation, in the first 15 min, the GA solution, in the absence or presence of $NO_3^-$ and Fe(III), was exposed to a stream of air (800 mL min$^{-1}$), followed by a 30 min exposure to a stream of 50 ppb of $NO_2$ (800 mL min$^{-1}$), then again exposed to a stream of air for 15 min. The spectral irradiance of the Xe lamp was measured with a calibrated spectroradiometer (Ocean Optics) equipped with a linear-array CCD detector (Supplementary Fig. S10).

**Estimations of the *DBE* and aromaticity equivalent ($X_C$) parameters**. The product compounds were further visualized by multiple structure-dependent tools to improve their identification and characterization[37] such as aromaticity equivalence ($X_C$), Kendrick mass defect (KMD) analyses, and Van Krevelen (VK) plots. The degree of unsaturation of organic compounds is described by the oxygen-to-carbon ratio (O/C) versus the hydrogen-to-carbon ratio (H/C) in Van Krevelen plot[40]. Here, we calculated the KMD values by normalizing the $CH_2$ group for CHO and CHON compounds according to the following Equations:

$$\text{KM} = \text{observed mass} \times \frac{\text{nominal mass of } CH_2}{\text{exact mass of } CH_2} \quad (11)$$

$$\text{KMD} = \text{nominal mass (NM)} - \text{KM} \quad (12)$$

The double bond equivalent (*DBE*) for elemental composition $C_cH_hO_oN_n$ can be expressed as follows:

$$DBE = 1 + \frac{1}{2}(2c - h + n) \quad (13)$$

The value of $X_C$, identifying the aromatic and condensed aromatic structures in a complex mixture of compounds, was expressed as follows:

$$X_c = \frac{2C + N - H - 2m \times O}{DBE - m \times O} + 1 \quad (14)$$

If $DBE \leq m \times o$ or $X_c \leq 0$, then $X_C = 0$, where $m$ is the fraction of O atoms in the π-bond structure of the compound. We used $m = 0.5$ for the calculation of $X_C$ in this study[37]. Threshold values of $X_C$ between 2.5 and 2.7 ($2.5 \leq X_C < 2.7$), indicating the presence of monoaromatic compounds, and equal or greater than 2.7 ($X_C \geq 2.7$), suggesting the presence of polyaromatic compounds, were set as minimum criteria for the presence of aromatic or condensed aromatic compounds in the identified

ions[37]. In our calculations, the values of $m$ and $n$ were set to 0.5 since the negative mode of ESI is most sensitive to compounds that contain carboxylic functional groups and nitro compounds[37,38].

**Long-path absorption photometer (LOPAP).** The mixing ratio of HONO, produced by the heterogeneous reaction of $NO_2$ with TA, TA/FeCl$_3$·6H$_2$O, TA/FeCl$_3$·6H$_2$O/NaNO$_3$, GA, GA/FeCl$_3$·6H$_2$O, GA/FeCl$_3$·6H$_2$O/NaNO$_3$, was online monitored in a long-path absorption cell after conversion into an azodye, absorbing at 550 nm. The long-path absorption cell is made out of Teflon tubing (Teflon AF2400), allowing light to be transferred in total reflection due to the low refractive index of the Teflon tubing. The detection limit was smaller than 3 ppt with a total accuracy of ±10% with an actual time response of about 5 min and a time resolution of 15 s.

## Data availability
Datasets for the research are available in Supplementary Data[35], and other data are available from the corresponding authors upon reasonable request.

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

## Acknowledgements

This work was financially supported by Ministry of Science and Technology of China (2022YFC3701102), the National Natural Science Foundation of China (42177087, 41977187, and 42207127), National Natural Science Foundation of China, Research Fund for International Scientists (4221101064), Chinese Academy of Science, International Cooperation Grant (132744KYSB20190007), State Key Laboratory of Organic Geo-chemistry, Guangzhou Institute of Geochemistry (SKLOG2020-5, and KTZ_17101), and Guangdong Foundation for Program of Science and Technology Research (2017B030314057), National Key R&D Program of China (2021YFC1808903), the Key-Area Research and Development Program of Guangdong Province (2019B110207002), Science and Technology Planning Project of Guangdong Province (200106105876892 and 190807115560881).

## Author contributions

Investigation: P.L., R.G., X.L., J.L., M.T., and X.W. Supervision: H.A.A. and S.G. Writing —original draft: P.L., S.G., and H.A.A. Writing—review & editing: P.L., S.G., Y.Y., and H.A.A. All authors discussed the results, contributed to the analysis, and commented on the manuscript.

## Competing interests

Hind A. Al-Abadleh is an Editorial Board Member for Communications Chemistry, but was not involved in the editorial review of, or the decision to publish this article. All other authors declare no competing interests.
