## [Peer Review File · Communications Chemistry]

Reviewers' comments:

Reviewer #1 (Remarks to the Author):

This work reports an increased variety of gas-phase N-containing compounds formed from heterogeneous oxidation of NO₂ with gallic acid in the presence of nitrate and iron in the solution, measured by secondary electrospray ionization mass spectrometry. Gallic acid was used as a proxy for biomass burning aerosol; nitrate and iron are ubiquitous in fog and cloud droplets. Thus, the authors imply that their results suggest that the number of N-containing compounds is significantly higher in deliquescent aerosol particles due to the influence of relatively higher ionic strength from nitrate ions and complexation/redox reactivity of iron cations. The results and analysis are interesting, but there are a few key problems with this work. First, the results are rather qualitative than quantitative. It is unclear whether they could directly improve our understanding of the heterogeneous oxidation process. A number of new products are mentioned with their O/C, N/C, Xc, and DBE, but how they are produced and the exact roles of nitrate and iron in the mechanism were not clearly discussed. Finally, using results from aqueous-phase reactions (using solutions) to infer aerosol processes always comes with a caveat that the process may not be the same as aerosol particles or droplets. Some additional comments are provided below.

1. Figure 1. The differences in intensities between zero air and samples are not very large. Where are the signals coming from for the zero air runs? I believe that the four m/z's shown are the better ones already. With this level of enhancement, the sensitivity of this analysis is likely very poor. Thus, it is possible that the many products were formed with NO₃ or iron but were not successfully detected. If another phenolic surrogate is used, could the many products be observed without NO₃ or iron?
2. The authors pointed out that some of the same formulas were observed in the field. But this only indicates the same formulas, not necessarily the same compounds. The structures and the mechanisms that lead to their formation should at least be proposed.
3. Concentrations are sometimes used μM and sometime mol L^{-1} . Please consider using a consistent unit.

Reviewer #2 (Remarks to the Author):

This manuscript describes the identification of gaseous products from the reactions of nitrogen dioxide with gallic acid, representing humic-like substances, in solutions mimicking deliquesced aerosol (i.e., high ionic strength nitrate and iron). Secondary electrospray ionization (in both positive and negative modes) ultra-high resolution mass spectrometry allowed for determination of molecular formulas for a broad array of gaseous products, the majority of which contained nitrogen functional groups. Control of experimental variables including the presence and concentration of nitrate and FeCl₃, and dark and irradiation conditions, provide insight into the mechanistic formation of the gaseous products. In summary, this experiments thoroughly introduce and model the complexity of real world conditions

and demonstrate the potential for these reactions to generate gaseous products of concern for air quality and the formation of light-absorbing brown carbon. In particular, several identified products match those found in the atmosphere, while others are known as highly toxic – making the results of this manuscript of significant interest to readers of Chemical Communications. I therefore recommend this manuscript for publication pending consideration of the following suggestions.

1. One notable result presented in this manuscript that is not discussed mechanistically is the observation that increasing nitrate concentrations under dark conditions significantly alter the number of gaseous products (e.g., S1). I encourage the authors to address this observation in relation to their thorough discussion of the impact of nitrate and iron on this chemistry. Is the reduction of gallic acid to quinones responsible for this dramatic loss in reactivity?
2. Nitrate also suppresses the release of HONO under light conditions in the presence of iron (e.g., Fig. S8 compared to Fig. S7), presumably due to absorption and photolysis of nitrate in the system. This observation is worthy of being discussed in the manuscript.

Minor edits:

3. The authors should define Double Bond Equivalency (BDE) when first presented (Fig. 2 and line 202).
4. Be consistent with structure notation in R-8 through R-11: GANO₂ vs. GA-NO₂ vs. GA – NO₂
5. Line 218: “O/C” ratio should be “N/C”
6. Figures S6 – S8 refer to NO₂ signal in each caption but do not include this data.

Reviewer #3 (Remarks to the Author):

General Comments. The manuscript by Li et al. reports findings from experiments on gas phase products formed when aq solutions of gallic acid are exposed to NO₂(g) in the presence and absence of nitrate and UV-visible irradiation. The main finding, and the one that is most novel is the use of SESI-UHR-MS to show that N-containing organic molecules are produced from the reported heterogeneous photochemical systems. These results suggest that N is becoming incorporated into the gallic acid substrate, ultimately forming N-containing volatile organic compounds. This is an interesting finding, although the authors don't discuss the mechanism for the appearance of these compounds (e.g. amines) and rather go into discussing the mechanism of HONO formation, which appears to be somewhat less novel based on the literature precedence. Seems like a missed opportunity.

The system that the authors chose is very complicated and because of this the results are very difficult to interpret. Indeed, the proposed mechanisms are based on likely scenarios where individual steps are supported by prior publications on simpler systems. Thus, I find there is a lot of speculation presented and the proposed mechanisms are not necessarily novel. It is a weakness of the paper that more supporting experiments could not be presented that support the proposed mechanisms. However, it could be argued that this is the nature of science, and one should be able to report the observations and leave it to the scientific community to decide whether the proposed mechanisms stand the test of time. Lastly, I would recommend that the authors carefully check the appropriateness of their references. Not all, but some seemed to not be from primary sources and/or did not seem to be related to the topic they were discussing.

Specific Comments (by line number)

(78-79) The authors state, "...oxidation of volatile organic compounds (VOCs) by atmospheric oxidants, especially NO₂, would lead to the formation of BrC.(ref 1)" This statement is inaccurate as BrC formation is initiated by ozone, hydroxyl radical, and nitrate radical. To make this statement correct, the authors would need to insert: "...oxidants, especially in the presence of NO₂, would lead to..."

(116, Scheme 1) The authors provide structures of catechol, syringol guaiacol, gentisic acid, and recorcinol, and use these structures to justify their use of Gallic acid for their present study. I don't quite understand the logic of this since it is suggesting that they chose gallic acid as a model for HULIS because other researchers chose these other compounds as model compounds of HULIS. Why not just say directly that gallic acid is a great model compound because it has the functional groups found in real HULIS, and then cite some papers that support this? This would make more sense in my opinion.

(132-134) It seems out of place to add experimental details in the introduction of the paper. Would be better to include these in the results section.

(162, Fig. 1) What are the peaks at m/z 91.9981, 103.0387, 117.0544, 131.0702. I presume you have calibrated the mass scale of your MS, so you can you assign structures to these and indicated them where they are first described?

(Fig 1) The use of MS peak intensity as a proxy for quantitative analysis is controversial since was no attempt to calibrate individual peak intensity. Certain compounds (e.g., CHON) will not ionize as well as CHO compounds, so the most you can do is count the formulae and show van Krevelen plots as you have done in Fig. 2 - 4

(Fig 2-3). Can the authors comment on the supporting work /control experiments they undertook to determine if the differences in the MS results in the presence and absence of nitrate is not due to some matrix effect. That is, that nitrate is not impacting the ionization efficiency of the various types of products.

(253) Reference #42 does not seem to be related to the toxicity of 1,4-diaminobutane. The paper does describes an assay looking at the toxicity of nanoparticles and phalocyanines. The latter are very different from aliphatic amines in their physiochemical properties.

(265) Please check this statement. Aliphatic nitrooxy groups do not absorb a whole lot of visible light, so this statement doesn't quite accurate. I was under the impression that the light absorbing properties of BrC are due to conjugation and presence of aromatic nitro groups, which do red-shift the UV-visible absorption spectra.

(336) The statement, "which means no OH radicals" seems to me to be inaccurate in light of reaction (R-6), which suggests OH is formed in the no-nitrate system as well. I apologise if I miss something here.

(355) The authors cite reference 18, but this is a review article. Please cite the primary source of this

information as it is unclear where these pH arguments are coming from.

(402) The authors state that the polyamine compounds detected as products are toxic. Please be careful about accuracy of such statements. While I am not sure about diphenylamine, but I am pretty sure that putrescine has very low toxicity, and yet the authors use putrescine as an example throughout of a toxic compound produced by their reaction. Please see: Til, H.P.; Falke, H.E.; Prinsen, M.K.; Willems, M.I. (1997). "Acute and subacute toxicity of tyramine, spermidine, spermine, putrescine and cadaverine in rats". *Food and Chemical Toxicology*. 35 (3–4): 337–348.

Experimental section: I felt it would have improved my understanding of the experiments set up if the authors would include a schematic of their reactor system.

Figure S9: It is not clear which emission spectrum is due to sunlight and the photolysis lamp. If the lamp is the solid line, then it does not overlap much with gallic acid, so this would not support that there is any photochemistry centered on gallic acid. This would undermine some of the mechanistic claims.

Point-by-point response to reviewers
Revision 1, June 21, 2023
COMMSCHEM-23-0258

Reviewer #1 (Remarks to the Author):

This work reports an increased variety of gas-phase N-containing compounds formed from heterogeneous oxidation of NO₂ with gallic acid in the presence of nitrate and iron in the solution, measured by secondary electrospray ionization mass spectrometry. Gallic acid was used as a proxy for biomass burning aerosol; nitrate and iron are ubiquitous in fog and cloud droplets. Thus, the authors imply that their results suggest that the number of N-containing compounds is significantly higher in deliquescent aerosol particles due to the influence of relatively higher ionic strength from nitrate ions and complexation/redox reactivity of iron cations.

The results and analysis are interesting, but there are a few key problems with this work. First, the results are rather qualitative than quantitative. It is unclear whether they could directly improve our understanding of the heterogeneous oxidation process.

Authors' response: The focus of this manuscript is on gas phase product identification from the heterogeneous reaction of NO₂ with a solution containing gallic acid (GA) as a proxy of HULIS at pH 5 relevant for moderately acidic aerosol particles. In the abstract, we state that “*Results showed that the number of detected N-containing organic compounds largely increased from 4 during the NO₂ reaction with GA in the absence of nitrate and iron ions to 55 in the presence of nitrate and iron ions.*” The technique we used is state of the art mass spectrometry that provides qualitative information on the products (masses and hence structural information) and quantitative information from the relative intensities. The literature summary we provided in the introduction clearly shows that there is a knowledge gap in this area. Hence, our manuscript will be the basis for future experimental work designed to obtain additional quantitative information such as absolute amounts of selected products and kinetics of reaction steps in the mechanism under the experimental conditions reported in the manuscript.

To address this comment, a new sentence was added to the last paragraph in the introduction on the advantages of the technique we used. Also, a new paragraph was added to the ‘Conclusions’ section to provide details on context and future directions.

A number of new products are mentioned with their O/C, N/C, Xc, and DBE, but how they are produced and the exact roles of nitrate and iron in the mechanism were not clearly discussed.

Authors' response: In the revised version of the manuscript, we added a new tentative reaction pathway (Scheme S3) to describe the formation of putrescine in the presence of Fe(III) and NO₃⁻ under dark conditions. The following detailed description of Scheme S3 was added to the Section on the “Role of Fe(III)”:

The N-containing compounds formed upon heterogeneous reaction of NO₂ with GA in the presence of Fe(III) in dark can be tentatively explained by formation of phenoxyl radicals (A₃)⁷⁷, which can

abstract H atom from hydroxy group of gallic acid (Scheme S3). Then A₃ undergoes nucleophilic substitution reaction in the presence of NO₃⁻ to yield 3,4-dihydroxy-5-nitrobenzoic acid (C₇H₅O₆N) (A₄)^{46,78,79}. A₄ can be reduced in the presence of Fe(III) to 5-amino-3,4-hydroxybenzoic acid (C₇H₇O₄N) (A₅)⁸⁰⁻⁸². A₅ can abstract H atom from hydroxy group to form a radical, followed by the nucleophilic substitution reaction to generate 4-hydroxy-5-amino-3-nitrobenzoic acid (C₇H₆O₅N₂) (A₇)^{46,77-79}. A₇ can be reduced in the presence of Fe(III) to 4-hydroxy-2,5-diaminobenzoic acid (C₇H₈O₃N₂) (A₈)⁸⁰⁻⁸². The A₈ could abstract H atom from hydroxy group by NO₂ radical to yield 2,5-diamino-1,4-benzoquinone (C₆H₆O₂N₂) (A₉)⁸³. Finally, the A₉ could be oxidized by OH radical to form (C₄H₆O₃N₂) (A₁₀)⁸⁴, which undergoes decarboxylation leading to putrescine (C₄H₁₂N₂) (A₁₁)⁸⁵ (Scheme S3).”

Finally, using results from aqueous-phase reactions (using solutions) to infer aerosol processes always comes with a caveat that the process may not be the same as aerosol particles or droplets.

Authors’ response: We thank the reviewer for their comment. The following sentences were added to the last paragraph in the Conclusions section:

“While bubbling NO₂ gas into an aqueous phase solution followed by analyzing gas phase products is not identical to flowing NO₂ over an aerosol particle or a suspended microdroplet, these experimental setups share the same elementary steps in the reaction mechanism such as adsorption, surface diffusion, accommodation to the bulk, and bulk diffusion (ref 11). Hence, results from the experimental setup described here provide a reference point and a bench mark for future work that aims to examine the roles of surface to volume ratio, particle curvature, and aerosol liquid water acidity and ionic strength.”

Ref 11: Ruiz-Lopez, M.F., Francisco, J.S., Martins-Costa, M.T.C. et al. Molecular reactions at aqueous interfaces. *Nat Rev Chem* 4, 459–475 (2020). <https://doi-org.proxy.lib.uwaterloo.ca/10.1038/s41570-020-0203-2>

Some additional comments are provided below.

1. Figure 1. The differences in intensities between zero air and samples are not very large. Where are the signals coming from for the zero air runs? I believe that the four m/z’s shown are the better ones already. With this level of enhancement, the sensitivity of this analysis is likely very poor. Thus, it is possible that the many products were formed with NO₃ or iron but were not successfully detected. If another phenolic surrogate is used, could the many products be observed without NO₃ or iron?

Authors’ response: Figure 1 shows absolute intensities recorded by the UHR-MS of the selected masses. Therefore, the signals coming from the zero-air stage are the overall appearance of a series of possible reasons. For example, the inherent background of the instrument signal, e.g., the signal output from the instrument when a blank is measured; generally, a voltage output that is digitized by an analog to digital converter resulting in some masses background signals not being zero. In addition, these noises can also come from electronic components, signal processing circuits,

electronic noise, etc., the algorithms in data processing and analysis may also lead to some masses background signals being non-zero values.

2. The authors pointed out that some of the same formulas were observed in the field. But this only indicates the same formulas, not necessarily the same compounds. The structures and the mechanisms that lead to their formation should at least be proposed.

Authors' response: Indeed, the same formulas do not imply that the same chemical structure as numerous structural isomers are probable for each chemical formula. Nevertheless, these results suggest that the effect of ionic strength and presence of iron ions on heterogeneous NO₂ processing on proxies of HULIS can affect the products distribution in aerosol particles. To address this comment, we suggested a tentative reaction mechanism (Scheme S3) describing the formation of putrescine in the presence of Fe(III) and NO₃⁻ under dark conditions. We added the following sentence to the last paragraph in the “Characterization of the gas-phase product compounds” section of the revised manuscript:

“It has to be noted however that the same chemical formula does not necessarily imply that it is the same molecular structure as number of structural isomers are plausible for each formula. (ref 14)”

Ref 14: Nizkorodov, S. A., Laskin, J. & Laskin, A. Molecular chemistry of organic aerosols through the application of high resolution mass spectrometry. *Phys. Chem. Chem. Phys.* **13**, 3612-3629 (2011).

3. Concentrations are sometimes used uM and sometime mol L⁻¹. Please consider using a consistent unit.

Authors' response: We thank the reviewer for their comment. The unit ‘uM’ was found in one location and replaced with mol L⁻¹ in the manuscript.

Reviewer #2 (Remarks to the Author):

This manuscript describes the identification of gaseous products from the reactions of nitrogen dioxide with gallic acid, representing humic-like substances, in solutions mimicking deliquesced aerosol (i.e., high ionic strength nitrate and iron). Secondary electrospray ionization (in both positive and negative modes) ultra-high resolution mass spectrometry allowed for determination of molecular formulas for a broad array of gaseous products, the majority of which contained nitrogen functional groups. Control of experimental variables including the presence and concentration of nitrate and FeCl₃, and dark and irradiation conditions, provide insight into the mechanistic formation of the gaseous products. In summary, this experiments thoroughly introduce and model the complexity of real world conditions and demonstrate the potential for these reactions to generate gaseous products of concern for air quality and the formation of light-absorbing brown carbon. In particular, several identified products match those found in the atmosphere, while others are known as highly toxic – making the results of this manuscript of significant interest to readers of Chemical Communications. I therefore recommend this manuscript for publication pending consideration of the following suggestions.

Authors' response: We thank the reviewer for their comment.

1. One notable result presented in this manuscript that is not discussed mechanistically is the observation that increasing nitrate concentrations under dark conditions significantly alter the number of gaseous products (e.g., S1). I encourage the authors to address this observation in relation to their thorough discussion of the impact of nitrate and iron on this chemistry. Is the reduction of gallic acid to quinones responsible for this dramatic loss in reactivity?

Authors' response:

Figure 1

Figure S1

We thank the reviewer for this comment. We believe the reviewer is referring to the data in Figures 1C and S1 above. The second paragraph in the 'Role of the nitrate ion' section provides the most likely mechanistic steps on the dark reaction of nitrate with gallic acid. To address this comment, the following sentences were added to that section:

“In addition, data in Figures 1C and S1 show that increasing nitrate concentrations under dark conditions significantly alter the number of gaseous products. Since GA is the limiting reagent in these redox paired reactions, it is very likely that the increase in ionic strength from increasing the nitrate concentration under dark conditions dramatically reduces the kinetics of subsequent oxidation pathways of GA following its initial oxidation to quinones. As a results, a limited number of products with $X_C < 2.5$ are observed at $[NO_3^-] = 0.5 M$ compared to the larger number of products with $2.5 \leq X_C < 2.71$ and $X_C > 2.71$ at $[NO_3^-] = 0.05 M$.”

2. Nitrate also suppresses the release of HONO under light conditions in the presence of iron (e.g., Fig. S8 compared to Fig. S7), presumably due to absorption and photolysis of nitrate in the system. This observation is worthy of being discussed in the manuscript.

Authors’ response:

Under irradiation, the levels of HONO formation in the presence of nitrate ($[NO_3^-] = 0.05 \text{ mol L}^{-1}$) (Figure S6) is the same as that in the presence of nitrate ($[NO_3^-] = 0.05 \text{ mol L}^{-1}$) and Fe(III) ($1 \times 10^{-6} \text{ mol L}^{-1}$) (Figure S8). However, HONO formed under irradiation in the presence of Fe(III) ($3 \times 10^{-5} \text{ mol L}^{-1}$) shown in Figure S7 is higher than that in Figure S8.

In the revised manuscript, the following reactions and sentence were added to the third paragraph in the “Role of Fe(III)” section:

“These reaction show that the Fe(II)/Fe(III) redox couple drives the formation of HONO. Gaseous NO_2 is reduced to NO_2^- by Fe(II) via an electron transfer reaction, followed by the protonation of NO_2^- yielding HONO.”

Minor edits:

3. The authors should define Double Bond Equivalency (BDE) when first presented (Fig. 2 and line 202).

Authors’ response: Done.

4. Be consistent with structure notation in R-8 through R-11: GANO2 vs. GA-NO₂ vs. GA – NO₂

Authors' response: Done.

5. Line 218: "O/C" ratio should be "N/C"

Authors' response: Done.

6. Figures S6 – S8 refer to NO₂ signal in each caption but do not include this data.

Authors' response: We removed the NO₂ signal from each caption in Figures S6 – S8.

Reviewer #3 (Remarks to the Author):

General Comments. The manuscript by Li et al. reports findings from experiments on gas phase products formed when aq solutions of gallic acid are exposed to NO₂(g) in the presence and absence of nitrate and UV-visible irradiation. The main finding, and the one that is most novel is the use of SESI-UHR-MS to show that N-containing organic molecules are produced from the reported heterogeneous photochemical systems. These results suggest that N is becoming incorporated into the gallic acid substrate, ultimately forming N-containing volatile organic compounds. This is an interesting finding, although the authors don't discuss the mechanism for the appearance of these compounds (e.g. amines) and rather go into discussing the mechanism of HONO formation, which appears to be somewhat less novel based on the literature precedence. Seems like a missed opportunity.

Authors' response: This comment is related to Comment #2 by Reviewer #1, which we addressed above on page 1. To avoid redundancy, please read our response above.

The system that the authors chose is very complicated and because of this the results are very difficult to interpret. Indeed, the proposed mechanisms are based on likely scenarios where individual steps are supported by prior publications on simpler systems. Thus, I find there is a lot of speculation presented and the proposed mechanisms are not necessarily novel. It is a weakness of the paper that more supporting experiments could not be presented that support the proposed mechanisms. However, it could be argued that this is the nature of science, and one should be able to report the observations and leave it to the scientific community to decide whether the proposed mechanisms stand the test of time. Lastly, I would recommend that the authors carefully check the appropriateness of their references. Not all, but some seemed to not be from primary sources and/or did not seem to be related to the topic they were discussing.

Authors' response: This comment is also related to Comment #2 and Additional Comment #2 by Reviewer #1, which we addressed above on page 1 and page 3. To avoid redundancy, please read our response above.

Specific Comments (by line number)

(78-79) The authors state, "...oxidation of volatile organic compounds (VOCs) by atmospheric oxidants, especially NO₂, would lead to the formation of BrC.(ref 1)" This statement is inaccurate as BrC formation is initiated by ozone, hydroxyl radical, and nitrate radical. To make this statement correct, the authors would need to insert: "...oxidants, especially in the presence of NO₂, would lead to..."

Authors' response: We thank the reviewer for this comment. The edit was made in the revised manuscript.

(116, Scheme 1) The authors provide structures of catechol, syringol, guaiacol, gentisic acid, and recorcinol, and use these structures to justify their use of Gallic acid for their present study. I don't quite understand the logic of this since it is suggesting that they chose gallic acid as a model for HULIS because other researchers chose these other compounds as model compounds of HULIS.

Why not just say directly that gallic acid is a great model compound because it has the functional groups found in real HULIS, and then cite some papers that support this? This would make more sense in my opinion.

Authors' response: We thank the reviewer for this comment. We kept Scheme 1 and made the suggested above in the revised manuscript and add the following reference:

Guangjie Zheng, Kebin He, Fengkui Duan, Yuan Cheng, Yongliang Ma, Measurement of humic-like substances in aerosols: A review, *Environmental Pollution*, Volume 181, 2013, Pages 301-314, <https://www.sciencedirect.com/science/article/abs/pii/S0269749113003151>

(132-134) It seems out of place to add experimental details in the introduction of the paper. Would be better to include these in the results section.

Authors' response: These sentences were edited by removing concentration values and emphasizing novel results.

(162, Fig. 1) What are the peaks at m/z 91.9981, 103.0387, 117.0544, 131.0702. I presume you have calibrated the mass scale of your MS, so you can you assign structures to these and indicated them where they are first described?

Authors' response: The following sentence was added to the caption of Figure 1:

“Table S1 lists the m/z 103.0387, 117.0544 and 131.0702 with tentative structures. The m/z 91.9981 corresponds to (Nitroperoxy)methane.”

(Fig 1) The use of MS peak intensity as a proxy for quantitative analysis is controversial since was no attempt to calibrate individual peak intensity. Certain compounds (e.g., CHON) will not ionize as well as CHO compounds, so the most you can do is count the formulae and show van Krevelen plots as you have done in Fig. 2 – 4

Authors' response: This comment is also related Additional Comment #2 by Reviewer #1, which we addressed above on page 1 and page 3. To avoid redundancy, please read our response above.

(Fig 2-3). Can the authors comment on the supporting work /control experiments they undertook to determine if the differences in the MS results in the presence and absence of nitrate is not due to some matrix effect. That is, that nitrate is not impacting the ionization efficiency of the various types of products.

Authors' response: Our blank experiments were done with stream of air in the absence of oxidants as shown in Figure 1D. The blank experiments with the stream of air flowing over the solution

made of GA, GA/NO₃⁻, GA/Fe(III), and GA/NO₃⁻/Fe(III) clearly showed that the nitrate ions do not affect the ionization efficiency.

(253) Reference #42 does not seem to be related to the toxicity of 1,4-diaminobutane. The paper does describes an assay looking at the toxicity of nanoparticles and phalocyanines. The latter are very different from aliphatic amines in their physiochemical properties.

Authors' response: We had removed the reference #42 and the following reference was cited in the revised manuscript:

Til, H. P., Falke, H. E., Prinsen, M. K. & Willems, M. I. Acute and subacute toxicity of tyramine, spermidine, spermine, putrescine and cadaverine in rats. *Food Chem. Toxicol.* **35**, 337-348 (1997).

(265) Please check this statement. Aliphatic nitrooxy groups do not absorb a whole lot of visible light, so this statement doesn't quite accurate. I was under the impression that the light absorbing properties of BrC are due to conjugation and presence of aromatic nitro groups, which do red-shift the UV-visible absorption spectra.

Authors' response: This sentence was edited for clarity and now reads “*These compounds with $X_C > 2.5$ exhibit $O/N = 4$ indicating they are aromatic compounds which contain nitrooxy ($-ONO_2$) group; hence, they could be light absorbing compounds associated with BrC in the atmosphere^{35,45}*”

(336) The statement, "which means no OH radicals" seems to me to be inaccurate in light of reaction (R-6), which suggests OH is formed in the no-nitrate system as well. I appogise if I miss something here.

Authors' response: We thank the reviewer for this comment. This sentence is a consequence of the previous three lines that highlight the formation of FeGA₃ complexes due to the presence of GA in excess of Fe(III). Hence, hydrated Fe(III) are not expected to be in solution that would photolyze to produce OH radicals. Hence, Reaction R-6 and R-10 were deleted and the reactions were renumbered in the revised manuscript.

(355) The authors cite reference 18, but this is a review article. Please cite the primary source of this information as it is unclear where these pH arguments are coming from.

Authors' response: We thank the reviewer for this comment. This section was edited, reference 18 deleted, and the following reference was added:

Gen, M.; Zhang, R.; Chan, C. K., Nitrite/nitrous acid generation from the reaction of nitrate and Fe(II) promoted by photolysis of iron–organic complexes. *Environ. Sci. Technol.* 2021, 55 (23), 15715-15723.

(402) The authors state that the polyamine compounds detected as products are toxic. Please be careful about accuracy of such statements. While I am not sure about diphenylamine, but I am pretty sure that putrescine has very low toxicity, and yet the authors use putrescine as an example throughout of a toxic compound produced by their reaction. Please see: Til, H.P.; Falke, H.E.; Prinsen, M.K.; Willems, M.I. (1997). "Acute and subacute toxicity of tyramine, spermidine, spermine, putrescine and cadaverine in rats". *Food and Chemical Toxicology*. 35 (3–4): 337–348.

Authors' response: We edited the statements, at the same time deleted the references 42, 74, 75, and added reference as follows:

For example, the polyamine compounds, such as 1,4-diaminobutane (putrescine), and aromatic amines containing two phenyl substituents, such as 4-dimethylaminobenzaldehyde, could be toxic compounds.

Til, H. P., Falke, H. E., Prinsen, M. K. & Willems, M. I. Acute and subacute toxicity of tyramine, spermidine, spermine, putrescine and cadaverine in rats. *Food Chem. Toxicol.* **35**, 337-348 (1997).

Walker, J. D. & Printup, H. Structure–Activity Relationships for Aldehyde Categories. *QSAR Comb. Sci.* **27**, 475-482 (2008).

Experimental section: I felt it would have improved my understanding of the experiments set up if the authors would include a schematic of their reactor system.

Authors' response:

We provided the illustration of the experimental set up in new Figure S9:

Figure S9: Simplified illustration of the experimental set up used in this study.

Figure S9: It is not clear which emission spectrum is due to sunlight and the photolysis lamp. If the lamp is the solid line, then it does not overlap much with gallic acid, so this would not support that there is any photochemistry centered on gallic acid. This would undermine some of the mechanistic claims.

Authors' response:

To address this comment, the revised manuscript shows the spectrum of the Xe lamp (black dashed line) in new Figure S10 (see below). This spectrum of the Xe lamp overlaps well with the absorption spectrum of GA and the mixtures of GA with NO_3^- and Fe(III).

Figure S10: The absorption spectra of GA ($1 \times 10^{-4} \text{ mol L}^{-1}$) in the presence of NaNO_3 ($I_{\text{eff}} = 0.05 \text{ M}, 0.5 \text{ M}$)/Fe(III) ($1 \times 10^{-6} \text{ mol L}^{-1}$), the photon flux of lamp (black dashed line). The left axis corresponds to the absorption spectra of GA in presence of NaNO_3 and Fe(III), and the right axis corresponds to the actinic flux of xenon lamp.

Reviewers' comments:

Reviewer #2 (Remarks to the Author):

The authors have responded to most of the reviewer comments to my satisfaction. The one area that still remains underdeveloped is a thorough understanding of the reaction mechanisms, with particular emphasis on the importance of each component of the mixture. The authors propose a "tentative mechanism" (Scheme S3) which begins to answer some of these aspects, but nonetheless remains speculative. The reduction of aromatic nitro groups to amines by iron processing provides a plausible mechanism critical to this system. However, the role of nitrate remains less clear. Scheme S3 shows nitrate ions as the mechanistic step for nitration of gallic acid through formation of nitrate esters. Bubbling NO₂ through aqueous solutions of phenols has been shown to generate nitrophenols, raising the question why these compounds are not formed in the absence of nitrate. Regardless, I still find the manuscripts worthy of publication as the identification of these products in this complicated system, which is a reasonable model for real-world chemistry, provides a stepping stone for future work to fully elucidate the kinetics and mechanisms of this chemistry.

Reviewer #4 (Remarks to the Author):

This paper seeks to identify products arising from the interactions of gas phase NO₂ with condensed phase molecule, gallic acid, that serves as a proxy for humic-like substances (HULIS). This type of system has traditionally been studied because HULIS was thought to be capable of (photo)reducing NO₂ and serving as a source of nitrite, with implications for the gas phase HONO budget. In this paper, the authors turn their attention to the N-containing organics that are emitted into the gas phase in their experimental system. In particular, they add high concentrations of nitrate and iron to incorporate some impacts from other redox-active constituents that are often present in atmospheric aerosol. A major and novel claim of the paper is that the presence of nitrate and iron in solution leads to the production of a larger number of organonitrogen compounds that evolve into the gas phase. Based on their molecular formulae and N/C and O/N ratios, the authors indicate that many of these compounds must represent reduced nitrogen. In the revised version of the paper, Scheme 3 proposes a multi-step pathway by which nitro groups are incorporated into the aromatic ring, reduced to amino groups and the ring is subsequently opened to produce an aliphatic diamine. Given the identification of reduced nitrogen in atmospheric brown carbon aerosol, and the potential health concerns posed by some forms of reduced N compounds, this is an interesting observation. The mechanism elucidated in Scheme 3 requires twelve atoms of Fe(II) to drive the reduction of nitro groups to amino groups, but iron is present at two orders of magnitude less than gallic acid in the system. Disappointingly, the section of the manuscripts that provides details on the impact of iron in the system focuses predominantly on the consequences of iron on the HONO production. The authors should be more explicit about whether any molecular formula to which they assigned the identity of reduced N were formed in the absence of iron. Additionally, the van Krevelen analysis is performed on some static representation of the gas phase products in the system. Based on the multi-step pathway described in Scheme 3, one might expect compounds with oxidized nitrogen to evolve from the system first, as the NO₂ is incorporated into the gallic acid, followed by a conversion to reduced nitrogen and

the NO₂ groups are converted to -NH₂ groups. Is there evidence from the high time resolution MS data that this is the case?

Do the authors have an explanation for massive increase in pH over the course of the experiments, and how that might impact the average conditions in the system when the heterogeneous chemistry was occurring?

Overall, the results of the study are intriguing, but they will be more influential to the environmental chemistry community if the role of nitrate and iron in the system can be understood more generally, so that the generalizability of the system is better understood.

Reviewer #2:

The authors have responded to most of the reviewer comments to my satisfaction. The one area that still remains underdeveloped is a thorough understanding of the reaction mechanisms, with particular emphasis on the importance of each component of the mixture. The authors propose a "tentative mechanism" (Scheme S3) which begins to answer some of these aspects, but nonetheless remains speculative. The reduction of aromatic nitro groups to amines by iron processing provides a plausible mechanism critical to this system. However, the role of nitrate remains less clear. Scheme S3 shows nitrate ions as the mechanistic step for nitration of gallic acid through formation of nitrate esters. Bubbling NO₂ through aqueous solutions of phenols has been shown to generate nitrophenols, raising the question why these compounds are not formed in the absence of nitrate. Regardless, I still find the manuscripts worthy of publication as the identification of these products in this complicated system, which is a reasonable model for real-world chemistry, provides a stepping stone for future work to fully elucidate the kinetics and mechanisms of this chemistry.

We are thankful for the constructive and encouraging comments. We did our best to describe the role of nitrate and iron ions on the reaction of NO₂ with Gallic acid. Surely, still remains further clarification to understand the dynamics of this complex system which is close to real-world.

The reviewer is right that the reaction of NO₂ with phenols in the aqueous phase can generate nitrophenols. However, we did not observe the formation of nitrophenols in the gas phase. We performed another study by using FTICRMS to detect and identify the aqueous phase products formed through the reaction of NO₂ with Gallic acid. In that study, that has been submitted elsewhere, we have detected nitrophenols as products formed during the reaction of NO₂ with Gallic acid which eventually could partition in the gas phase. The later depends on the volatility of these compounds. Most likely our experimental setup and/or the sensitivity of SESI-UHR-MS in this study did not allow the detection of nitrophenols in the gas phase.

Reviewer #4

This paper seeks to identify products arising from the interactions of gas phase NO₂ with condensed phase molecule, gallic acid, that serves as a proxy for humic-like substances (HULIS). This type of system has traditionally been studied because HULIS was thought to be capable of (photo)reducing NO₂ and serving as a source of nitrite, with implications for the gas phase HONO budget. In this paper, the authors turn their attention to the N-containing organics that are emitted into the gas phase in their experimental system. In particular, they add high concentrations of nitrate and iron to incorporate some impacts

from other redox-active constituents that are often present in atmospheric aerosol. A major and novel claim of the paper is that the presence of nitrate and iron in solution leads to the production of a larger number of organonitrogen compounds that evolve into the gas phase. Based on their molecular formulae and N/C and O/N ratios, the authors indicate that many of these compounds must represent reduced nitrogen. In the revised version of the paper, Scheme 3 proposes a multi-step pathway by which nitro groups are incorporated into the aromatic ring, reduced to amino groups and the ring is subsequently opened to produce an aliphatic diamine. Given the identification of reduced nitrogen in atmospheric brown carbon aerosol, and the potential health concerns posed by some forms of reduced N compounds, this is an interesting observation.

We are thankful for the encouraging comments.

The mechanism elucidated in Scheme 3 requires twelve atoms of Fe(II) to drive the reduction of nitro groups to amino groups, but iron is present at two orders of magnitude less than gallic acid in the system.

We had revised the Scheme S3, which shows the Fe(III)/Fe(II) cycle in the heterogeneous reaction of NO₂ with Gallic acid in the presence of Fe(III) and nitrate.

In the core of the manuscript we added the following:

“The N-containing compounds formed upon heterogeneous reaction of NO₂ with GA in the presence of Fe(III) in dark can be tentatively explained by formation of GA₃Fe complexes (A₁)⁶⁴, which reacts with NO₂ to generate Fe(II) and phenoxy radical (A₂). A₂ obtains an electron and an H atom to form phenoxy radicals (A₃)⁶⁷, which can abstract H atom from hydroxy group of GA (Scheme S3).”

Disappointingly, the section of the manuscripts that provides details on the impact of iron in the system focuses predominantly on the consequences of iron on the HONO production. The authors should be more explicit about whether the any molecular formula to which they assigned the identity of reduced N were formed in the absence of iron. Additionally, the van Krevelen analysis is performed on some static representation of the gas phase products in the system. Based on the multi-step pathway described in Scheme 3, one might expect compounds with oxidized nitrogen to evolve from the system first, as the NO₂ is incorporated into the gallic acid, followed by a conversion to reduced nitrogen and the NO₂ groups are converted to -NH₂ groups. Is there evidence from the high time resolution MS data that this is the case?

We performed real-time measurements of the formed gas phase product compounds under the following experimental conditions: 1) heterogeneous reaction of NO₂ with GA (Table S3), 2) GA/NO₃⁻ [NO₃⁻] = 0.05 mol L⁻¹ (Table S7), and 3) GA/NO₃⁻ [NO₃⁻] = 0.5 mol L⁻¹ (Table S9) in dark and, 4) heterogeneous reaction of NO₂ with GA (Table S4), 5) GA/NO₃⁻ [NO₃⁻] = 0.05 mol L⁻¹ (Table S8), and 6) GA/NO₃⁻ [NO₃⁻] = 0.5 mol L⁻¹

¹, under light irradiation (Table S10). After identifying all molecular formula, we did not find any reduced N in the absence of iron.

Do the authors have an explanation for massive increase in pH over the course of the experiments, and how that might impact the average conditions in the system when the heterogeneous chemistry was occurring?

The pKa value of GA is influenced by the variation of nitrate ions concentration that affects the acid-base equilibrium and hence modifies its protonation/deprotonation degree and, consequently, its UV-VIS absorption spectrum (Figure S10). Figure S10 shows that the addition of nitrate ions causes both the hypsochromic effect and the bathochromic shift. This effect i.e. the increase of pH with addition of inorganic ions has been previously observed during the reactions of ozone with methoxyphenols (Mekic et al., 2020, Wang et al., 2021)

Mekic, et al., Ionic strength effect alters the heterogeneous ozone oxidation of methoxyphenols in going from cloud droplets to aerosol deliquescent particles, *Environ. Sci. Technol.*, 2020, 54, 20, 12898-12907.

Wang, et al., Ionic Strength Effect Triggers the Brown Carbon Formation through Heterogeneous Ozone Processing of o-Vanillin, *Environ. Sci. Technol.*, 2021, 55, 8, 4553–4564.

Overall, the results of the study are intriguing, but they will be more influential to the environmental chemistry community if the role of nitrate and iron in the system can be understood more generally, so that the generalizability of the system is better understood.

We did our best to describe the role of nitrate and iron ions on the reaction of NO₂ with Gallic acid. Surely, still remains further clarification to understand the dynamics of this complex system which is close to real-world.

REVIEWERS' COMMENTS:

Reviewer #4 (Remarks to the Author):

I think the authors have done a reasonable job in interpreting the information they have available from the experiments and that the manuscript can be considered for publication in its most recent form.